# Association between Mutuality and Health-Related Quality of Life in Patient–Caregiver Dyads Living with Schizophrenia

**DOI:** 10.3390/ijerph18052438

**Published:** 2021-03-02

**Authors:** Chiu-Yueh Hsiao, Huei-Lan Lu, Yun-Fang Tsai

**Affiliations:** 1School of Nursing, College of Medicine, Chang Gung University, Taoyuan City 333, Taiwan; koalahsiao@gmail.com; 2Department of Psychiatry, Chang Gung Memorial Hospital, Taoyuan City 333, Taiwan; 3Director of Nursing Department, Jianan Psychiatric Center, Ministry of Health and Welfare, Rende, Tainan 717, Taiwan; 14404@mail.cnpc.gov.tw; 4Department of Psychiatry, Chang Gung Memorial Hospital in Keelung, Keelung City 204, Taiwan

**Keywords:** mutuality, health-related quality of life, schizophrenia, patient–caregiver dyads, actor–partner interdependence model

## Abstract

Background: Although caregivers are a crucial support in the recovery of patients with schizophrenia, little is known about how mutuality is related to health-related quality of life within the dyadic (patient and caregiver) context. This study aimed to investigate the dyadic relationship between mutuality and health-related quality of life in patients with schizophrenia and caregivers. Methods: A cross-sectional, correlational study was conducted with a sample of 133 dyads of patients with schizophrenia and caregivers. Structured questionnaires were used to collect data. Data were analyzed using descriptive statistics, paired sample *t*-tests, Pearson’s product-moment correlations, and the Actor–Partner Interdependence Model. Results: Mutuality of patients was significantly higher than that of caregivers. Compared with caregivers, patients had significantly lower total scores for health-related quality of life. Patients’ and caregivers’ mutuality was related to their own health-related quality of life (actor effect) and their partners’ health-related quality of life (partner effect). Conclusion: Mutuality plays a critical role in health-related quality of life in dyads of patients with schizophrenia and caregivers. Viewing a dyad as a unit of nursing care reveals a promising approach for developing recovery-oriented modalities targeted at stimulating mutuality that may enhance health-related quality of life for both patients and caregivers.

## 1. Introduction

Recovery is recognized as an ultimate goal for mental healthcare services [1]. Worldwide, the disability-adjusted life years of schizophrenia increased by 62.46% in 2017 compared to 1990 [2]. Inpatient psychiatric rehabilitation as an interim pathway provides recovery-oriented care to enable people who are disabled by mental disorders to live satisfactory lives [3,4]. Approximately 10–20% of individuals with complex forms of psychosis receive care in inpatient psychiatric rehabilitation treatment before community reintegration [4]. Health-related quality of life (HRQoL) in persons with schizophrenia is a measure of the integration of an individual’s mental illness with their social and environmental features beyond a reduction in symptoms alone [5,6]. Following inpatient psychiatric rehabilitation care, most patients with schizophrenia experience improvement in HRQoL [3] and success in sustained community living [4]. Noticeably, the unique bond that exists between individuals (patients and caregivers) in a close relationship impacts each other’s experiences [7]. With an increased focus on holistic care in mental health practice, it is crucial to understand correlates of HRQoL in patient–caregiver dyads facing schizophrenia.

Socio-demographics (age, gender, income) and clinical characteristics (illness duration) have been shown to be moderately associated with HRQoL in patients diagnosed with schizophrenia [8]. Variables relevant to HRQoL in caregivers include socio-demographics of both patients and caregivers [9,10,11] combined with the patients’ duration of mental illness and severity of psychiatric symptoms [9,11]. Furthermore, social support and coping have been reported as significant in promoting HRQoL in patients with schizophrenia [12] and their caregivers [10]. These results suggest that the interactions between patients and caregivers are critical to HRQoL within the dyadic relationship.

Mutuality is the degree of positivity in caregiver–care recipient relationships [13]. A systematic review found mutuality was correlated with positive outcomes for caregivers caring for frail older adults and patients with chronic diseases [14]. Greater mutuality was associated with higher HRQoL in outpatients suffering from schizophrenia [15] and caregivers of inpatients diagnosed with schizophrenia [16]. Some studies have investigated the association between mutuality and HRQoL from dyadic views of patients and their caregivers facing Parkinson’s disease [17] and heart failure [18]. However, no dyadic research has examined the relation of mutuality with HRQoL in patients with mental illness and caregivers.

The Actor–Partner Interdependence Model (APIM) [19] proposes the interdependence in dyadic relationships that simultaneously estimates the degree to which an individual’s independent variables influence his or her own outcome variables (i.e., actor effect) and his or her partner’s outcome variables (i.e., partner effect). The APIM has been adopted in the context of patients and caregivers coping with heart failure [18,20] to understand the dyadic effects of mutuality on HRQoL and psychological problems (e.g., anxiety and depression). However, no research has applied a dyadic approach (e.g., APIM) to examine the interdependence of patient–caregiver dyads in the face of mental health difficulties such as schizophrenia.

Earlier studies on patients receiving inpatient psychiatric rehabilitation services primarily investigated clinical outcomes of social functioning and costs for patients [4,21] rather than addressing the more subjective metric of HRQoL. Most aforementioned research has investigated the relationship of patients’ and caregivers’ characteristics with the HRQoL of patients with schizophrenia [8] and caregivers [9,10,11] from individual perspectives. This individualized approach ignores the interdependence within patient–caregiver dyads. In addition, these factors of HRQoL were mainly on an individual level. Other dyadic attributes (e.g., mutuality) of HRQoL for patients and caregivers have yet to be explored.

Considering the literature on patient–caregiver dyads in medical diseases, one would anticipate that a reciprocal influence exists within the dyadic context of recovery for patients with mental disorders such as schizophrenia. However, there is a gap in the literature regarding the dyadic relationship of mutuality with HRQoL in individuals suffering from schizophrenia and their caregivers. This gap poses a major barrier to identifying risks for patient–caregiver dyads, which prevents an advanced understanding of the dyadic nature and the development of strategies for improving HRQoL within patient–caregiver dyads. Further, research on patients with persistent and complicated demands in inpatient psychiatric rehabilitation facilities is limited [4]. Therefore, the aims of this study were to investigate the dyadic relationship between mutuality and HRQoL in patients with schizophrenia and caregivers. It was hypothesized that higher levels of mutuality would be related to better HRQoL in patient–caregiver dyads confronting schizophrenia.

## 2. Materials and Methods

### 2.1. Study Design and Sampling

This was a cross-sectional, correlational study following the Strengthening the Reporting of Observational Studies in Epidemiology (STROBE) standards [22]. A convenience sample of 133 dyads of patients with schizophrenia and caregivers was recruited from the inpatient psychiatric rehabilitation facilities of two psychiatric hospitals in Taiwan. Inpatient psychiatric rehabilitation facilities with recovery-oriented practice provide 24 h mental health support for patients who require longer inpatient treatment and are not yet ready for successful community reintegration.

Inclusion criteria of patients were as follows: (a) diagnosed with schizophrenia based on the criteria of the Diagnostic and Statistical Manual of Mental Disorders, Fifth Edition (DSM-5) [23]; (b) residing in an inpatient psychiatric rehabilitation facility for at least six months at recruitment; (c) age of 20 years or older; and (d) capable of understanding or communicating in Taiwanese or Mandarin. Caregivers were selected if they were (a) the primary person managing the patient’s mental disorder; (b) identified by patients as their caregivers; (c) age of at least 20 years; and (d) capable of understanding or communicating in Taiwanese or Mandarin. Both patients and caregivers consented to participate in this study.

A statistical power dyadic analysis was performed based on multiple linear regression [24]. The required sample size was 114 dyads, applying G*Power for the *F* test and multiple linear regression to achieve a power of 80% with a medium effect (R^2^ of 0.13 and *f*^2^ of 0.15) at a confidence level of 0.05.

### 2.2. Ethical Approval

The study was approved by the institutional review boards (IRBs) of Chung Shan Medical University Hospital (No. CS13257) and Jianan Psychiatric Center, Ministry of Health and Welfare (No. 14-021) prior to initiating this study. The purpose and procedures of the current study were explained to potential dyads. Participants were reassured that data would be kept confidential and participation was voluntary. All patient–caregiver dyads signed written informed consent forms prior to collecting data.

### 2.3. Data Collection

All patient–caregiver dyads who met the inclusion criteria and signed consent forms to participate were interviewed by a trained research assistant (RA). Structured questionnaires were conducted separately to obtain the demographic and measurement data described above.

### 2.4. Measures

#### 2.4.1. Demographic Information Sheet

Caregivers’ sociodemographic characteristics included age, gender, relationship to the patient, marital status, occupation, education, and monthly income. Patients’ sociodemographic characteristics (i.e., age, gender, marital status, occupation, and education) and clinical features (i.e., length of mental illness and number of psychiatric hospitalizations) were obtained from the patients’ charts.

#### 2.4.2. Chinese Version of the 18-Item Brief Psychiatric Rating Scale

The Chinese version of the 18-item Brief Psychiatric Rating Scale (BPRS) [25] was used to measure the presence and severity of psychiatric symptoms entailing positive symptoms, general psychopathology, and affective symptoms (e.g., thought disturbance, emotional withdrawal, hostility, and suspiciousness) for patients with mental illness, particularly schizophrenia. The first 16 items were rated with a 7-point Likert scale, and the last two items were scored on a 5-point Likert scale, where 0 reflects no symptoms and 7 (or 4) represents extremely severe symptoms. Higher scores indicated more severity of psychiatric symptoms. A trained RA conducted the assessments. For this study, Cronbach’s alpha was 0.78.

#### 2.4.3. Chinese Version of Mutuality Scale

Mutuality was assessed with the Chinese version of the Mutuality Scale, which consisted of questions related to love, shared activities, values, and reciprocity [26]. Each item was rated with a 5-point Likert scale ranging from 0 (“not at all”) to 4 (“a great deal”); the total score was the mean of all items (ranging from 0 to 4). Higher scores implied greater mutuality. In the current study, Cronbach’s alpha was 0.96 for patients and 0.94 for caregivers, respectively.

#### 2.4.4. Taiwanese Version of the 28-Item World Health Organization Quality of Life-Brief Form

HRQoL of the dyads was measured with the Taiwanese version of the 28-item World Health Organization Quality of Life-Brief Form (WHOQOL-BREF) [27], which included four aspects: physical health, psychological health, social relationships, and environment. Each item was scored on a 5-point Likert scale. The total scores ranged from 28 to 140; each domain score ranged from 4 to 20. A higher score revealed better HRQoL. Cronbach’s alpha of the total scale ranged from 0.91 to 0.94 across different samples [27]. In this study, the Cronbach’s alpha for patients was 0.92 and 0.94 for caregivers.

### 2.5. Statistical Analysis

Data were analyzed using SPSS software package Version 22 (SPSS, Chicago, IL, USA). Descriptive statistics were performed to characterize sociodemographic and clinical characteristics. Paired sample *t*-tests were computed to compare mutuality and HRQoL and its domains between patient–caregiver dyads. Pearson’s product-moment correlation was applied to identify the association of mutuality and HRQoL and its domains between patients and caregivers. In this study, the Actor–Partner Interdependence Model (APIM) [19] examined the association of patients’ and caregivers’ mutuality (independent variables) with their own and their partners’ HRQoL and its domains (outcome variables). The actor effect was the effect of an individual’s mutuality on his/her own HRQoL and its domains. The partner effect was the impact of an individual’s mutuality on his/her partner’s HRQoL and its domains. To clarify the understanding of the data analysis, the partner, hereafter, is referred to as a caregiver of a patient with schizophrenia. The presence of a partner effect indicated patients and caregivers as partners were nested within a dyad. The APIM analysis was performed by the online application APIM_SEM, developed by Stas and colleagues [28], which applied structural equation modeling with full information maximum likelihood (FIML) estimation using the lavaan software package, which ensured all available data were analyzed. The APIM accounted for individual and dyadic effects by estimating the influence of both individuals in a dyad on each other [19].

Prior to the APIM analysis, individual data were converted to dyadic-structure data. Covariates added to the APIM were selected from previous research [8,9,10], which included the patient’s age, gender, length of mental illness, and severity of psychiatric symptoms and the caregiver’s age, gender, and monthly income. The patients’ and caregivers’ independent variables (i.e., mutuality) and covariates were grand-mean centered based on the recommendations of Kenny et al. [19]. Assumptions of normality were met. In the case of missing data, pairwise deletion was applied. A two-tailed significant level of *p* < 0.05 was applied to all analyses.

## 3. Results

Characteristics of patient–caregiver dyads are presented in Appendix A. As displayed in Appendix A, patients reported a higher mean score for the Mutuality Scale than that of caregivers (*t* = 5.6, *p* < 0.001). Mean scores of HRQoL and its domains in patients were significantly lower than those in caregivers, with the exception of the psychological domain. Additionally, significant correlations were identified between patients’ mutuality and caregivers’ mutuality (*r* = 0.59, *p* < 0.001) as well as mean scores of HRQoL and its domains for patients and caregivers, with coefficients ranging from 0.26 to 0.56.

Figure 1 depicts that patients’ and caregivers’ greater mutuality were significantly related to better HRQoL for themselves (actor effects) and their partners (i.e., caregivers and patients) (partner effects).

As presented in Figure 2, significant sole partner effects of mutuality on physical health for patients and caregivers were found. Actor effects of patients’ and caregivers’ mutuality on their own psychological health (β = 0.28, *p* < 0.01 for patients; β = 0.36, *p* < 0.001 for caregivers) and the partner effects of caregivers’ mutuality on patients’ psychological health (β = 0.21, *p* < 0.05) were detected. Actor and partner effects of mutuality on social relationships for patients and caregivers were significant. Actor effects of patients’ mutuality and caregivers’ mutuality on their environment (β = 0.37, *p* < 0.001 for patients; β = 0.43, *p* < 0.001 for caregivers) and a partner effect of mutuality on the environment from caregivers to patients (β = 0.3, *p* < 0.001) were found.

## 4. Discussion

### 4.1. Innovation of the Research

This is the first study, to the best of our knowledge, to demonstrate a dyadic effect of mutuality on HRQoL in patients with schizophrenia in inpatient psychiatric rehabilitation facilities and their caregivers. The adoption of an APIM approach in the present study views patients and caregivers as interacting as a unit and expands our understanding of the dyadic nature by estimating the association of mutuality with an individual’s own HRQoL and the other individual’s HRQoL. Our main findings showed there were both actor and partner effects of mutuality on HRQoL in patient–caregiver dyads. Patients’ and caregivers’ mutuality were related not only to their own HRQoL but also to HRQoL of the other individual of the dyad.

Patients and caregivers in our study were middle-aged. In line with earlier research [9,29], most of the patients were unmarried and unemployed. In this study, the length of mental illness was 16.42 years, and the severity of psychiatric symptoms measured by the BPRS was 8.48, indicating schizophrenia appeared to be a persistent illness and its symptoms for patients were stabilized. Taking into account inpatient psychiatric rehabilitation settings and the chronic nature of schizophrenia, further exploration into attributes of recovery would be informative to target effective modalities to help patients reintegrate into the community with the least amount of help from healthcare providers.

In the present study, mutuality was significantly higher in patients than in caregivers, which is comparable with previous research for patients with heart failure [20]. This could reflect a greater sense of patients’ dependence on caregivers or could imply that caregivers’ perceptions of mutuality with patients gradually declined over the course of schizophrenia. There were significant differences between patients and caregivers in HRQoL except in the psychological aspect. Patients’ self-assessments were significantly lower than those of caregivers. This result contrasts with prior research indicating the HRQoL of Chinese patients with severe psychological disorders residing in psychiatric hospitals and communities was similar to that of their caregivers [30]. The disparities between the two studies could be explained by differences in patients’ diagnoses or the unequal numbers of patients and caregivers in the study by Guan and colleagues [30]. In addition, this study applied paired sample *t*-tests to compare the HRQoL of patients and caregivers as dyads using the same measure of the Taiwanese version of the 28-item WHOQOL-BREF for patients and caregivers.

Killaspy et al. [4] postulated that patients receiving care in inpatient psychiatric rehabilitation facilities often present complex mental health needs. It is possible that the mental health care needs of the patients in our study were not being met, and thus they perceived a lower HRQoL [31]. Additional work on service needs and HRQoL of patients in recovery is required. Responses are likely to affect and be affected within an interpersonal relationship of two intimate individuals [32]. This suggests that the psychological aspect of HRQoL in dealing with schizophrenia may be relevant to both patients and caregivers. Little is known about the comparison of HRQoL and its domains in patients suffering from schizophrenia and their caregivers. Further work is required to investigate the trajectory of schizophrenia in mutuality and HRQoL within patient–caregiver dyads.

Mutuality of patients and caregivers in our study was significantly correlated, which is in line with dyadic relationships for patients diagnosed with Parkinson’s disease [17] and heart failure [18]. Our findings also demonstrated a relationship between HRQoL and its domains between patients and caregivers. Similarly, Caqueo-Urízar et al. [29] asserted that caregivers’ HRQoL is a major concern for the improvement of HRQoL in patients with schizophrenia. Given that patients and caregivers are embedded in an interpersonal relationship, we suggest that mutuality and HRQoL in patient–caregiver dyads are not independent.

Regarding actor effects, mutuality of patients and caregivers was positively related to their own HRQoL and all of the domains except physical health. Our results for mutuality are similar to findings for patients with heart failure and their caregivers, which were related to their own self-care confidence [18]. In this study, neither patients’ mutuality nor caregivers’ mutuality was correlated with their own physical components of HRQoL, which is in contrast to prior research indicating that higher mutuality was associated with their own improved HRQoL [17,33]. Notably, the effects of mutuality on HRQoL in these studies were examined from individual perspectives rather than viewing both individuals of the dyads as a unit of analysis. Additional dyadic research examining actor effects of mutuality on HRQoL across domains is required.

Partner effects were identified in the association between mutuality and HRQoL. Greater patients’ mutuality was correlated with better caregivers’ HRQoL and its physical and social domains. Particularly, greater mutuality of caregivers was related to better HRQoL in all domains in patients, suggesting that when caregivers expressed more positive mutuality with patients, patients perceived better HRQoL. One could argue that caregivers’ mutuality is more likely to have a substantial relationship with the HRQoL of patients. Therefore, involving caregivers in patients’ mental health care is recommended to improve HRQoL for patients and caregivers. Nonetheless, no research has examined the dyadic dynamics in the context of schizophrenia or other mental illnesses, thus restricting further comparison of our findings. Empirical work on the correlation between mutuality and HRQoL in dyads of patients with schizophrenia and other mental illnesses and caregivers is warranted.

### 4.2. Implications

Patients and caregivers are naturally interdependent dyads in which schizophrenia affects not only patients but also caregivers. Our findings shed light on the clinical implications pertaining to individual members of a dyad as well as the dyad as a unit. Mental healthcare providers should understand the dynamics of dyads and involve both patients and caregivers as partners in mental healthcare rehabilitation. Assessment of mutuality in both patients and caregivers is particularly critical to detect patient–caregiver dyads at high risk of poor HRQoL. Despite the fact that patient–caregiver dyads may experience similar challenges, they also have their own unique needs and responses that should be acknowledged and supported by mental healthcare providers. Psychosocial interventions with individual- and dyadic-oriented approaches can help facilitate patients and caregivers to open dialogues about their desires, set realistic expectations, enhance reciprocity of sentiment, and appreciate efforts made for one another’s situations, which, in turn, results in improvement in HRQoL for patients and their caregivers [34]. Furthermore, recovery-oriented mental health practice should be highly recommended at a political level by implementing tailored interventions engaging both patients and caregivers as a dyad. Subsequently, these interventions may help dyads collaborate in ways that effectively manage mental health difficulties and enhance HRQoL in patient–caregiver dyads.

### 4.3. Limitations

There are several limitations to this study. Given a cross-sectional design, findings from the current study preclude understanding the causal relationships between mutuality and HRQoL for participant dyads during inpatient psychiatric rehabilitation services. Longitudinal research is recommended to confirm our findings. Other potential factors, such as length of admission in inpatient psychiatric rehabilitation facilities at recruitment or treatment duration of patients relevant to HRQoL, warrant further investigation. Lastly, results found within our group may not be generalized to other clinical settings, such as outpatient clinics.

## 5. Conclusions

Schizophrenia is a challenging and costly mental health problem for patients that has a reciprocal influence on caregivers. Differences between patients and caregivers in mutuality and HRQoL were found in this study. Specifically, mutuality is a key facet of improving HRQoL in patient–caregiver dyads. Mental healthcare providers should empower patients and caregivers to work together toward establishing mutually supportive partnerships to promote HRQoL for individual members and dyads in recovery-oriented psychiatric rehabilitation services.

## Figures and Tables

**Figure 1 ijerph-18-02438-f001:**
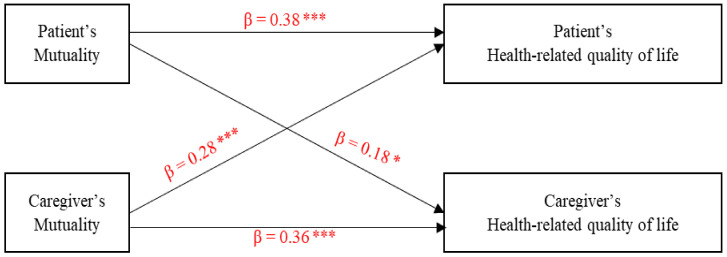
Actor and partner effects of mutuality on health-related quality of life. * *p* < 0.05, *** *p* < 0.001.

**Figure 2 ijerph-18-02438-f002:**
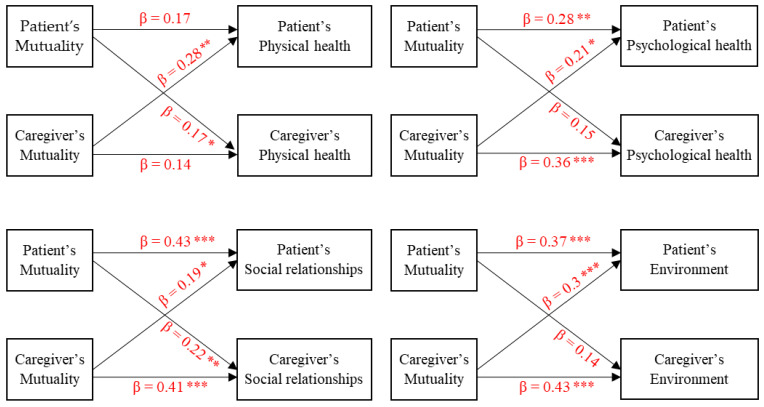
Actor and partner effects of mutuality on the domains of health-related quality of life (i.e., physical health, psychological health, social relationships, and environment). * *p* < 0.05, ** *p* < 0.01, *** *p* < 0.001.

## Data Availability

The data presented in this study are available on request due to privacy restrictions.

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
