# Peer review of "Association between Mutuality and Health-Related Quality of Life in Patient–Caregiver Dyads Living with Schizophrenia"

_ijerph, 2021, doi:10.3390/ijerph18052438_

Round 1

Reviewer 1 Report

Overall a very sound and well conducted study mainly based on the APIM model which brings significant innovation to the study of both direct and cross effects of the studied variables on patients and their partners.

The study is very well and concisely described: a clear and focused introduction; a sound methodological description; canonical presentation of results; well balanced discussion of results pointing to what can be extracted from the results but also to the need for more studies to understand some contradictory (taking into account other literature results) results. Indeed, as the authors point out, longitudinal studies would be welcomed to gain a more processual perspective on the ongoing dyadic relationships and their impact on health indicators.

I would just point to two minor change suggestions:

Page 6, immediately after Figure 2 (first line of the following paragraph): where it says "The only significant partner effects" I would suggest saying "The only significant sole partner effects".

Page 8, line 5: Where it says "Patient's mutuality was correlated to with...", it should say "Patient's mutuality was correlated with…" (omission of the "to").

Reviewer 2 Report

Association between mutuality and health-related quality of life in patient-caregiver dyads living with schizophrenia

I appreciated a wonderful opportunity to review the important issue in people with schizophrenia and their caregivers. The topic is important, but some areas needed to be improved.

Major revision:

  1. Demographic variables: In the materials and methods section, the demographic and clinical questions asked to the patients and caregivers and their categorization in the Table 1 has to be clearly mentioned in the Supplementary file. Only the variable names are listed in the main text which is not sufficient. A clear description of the variables are needed.

Minor revisions:

  1. The alignment of the text in the PDF has to be checked. The first and second sentences of multiple paragraphs need to be aligned.
  2. The word “also” has to be changed to “In addition” throughout the manuscript.
  3. Introduction, second line, include a space between approximately and 10-20%.
  4. 2. Use the word "Mutuality Scale" instead of the abbreviation.
  5. Check the alignment of figures under the results heading.
  6. The references has multiple spacings and has to be aligned.

Reviewer 3 Report

The topic of the manuscript warrants attention and might be of interest to readers. The present version of the manuscript requires - in my view - some improvements. These are as follows: 1) in the Introduction some more information concerning mutuality and its role in schizophrenia and recovery form could be provided; 2) it is not clear from the presentation why as participants those who resided in the inpatient facility were selected and for how long they resided in such facility before recruitment - for such selection of participants it is problematic in which context they put their mutuality and relationship with caregivers, what is more their HRQoL might be affected by their "inpatient status". Such situation is not addressed by the Authors; 3) in the presentation of the results repetition of data in tabels/figures and text could be omitted.  Although (1) and (3) can easily been corrected but (2) is related to the design of the study and needs to be clarified. 

Author Response

Response to Reviewer 3

We would like to express our gratitude and appreciation to you for the time and effort to review our manuscript. We appreciate the thoughtful suggestions and comments to help improve our manuscript. We have tried our best to revise address the concerns you raised. Changes in the text have been revised in red. Specific revisions are shown below.  

Regarding the rating points of the manuscript, the introduction included relevant references must be improved as well as research design, methods, and results can be improved.

Reply:

  1. Regarding the introduction included relevant references, we have added more information related to mutuality in schizophrenia and recovery in the introduction. Please see pages 1-2 and references.
  1. As for research design and methods, we added more information about inpatient psychiatric rehabilitation and inclusion criteria of patients in the study design and sampling. Please see page 3.
  1. In relation to results, we have condensed the results to avoid the repetition of data in tables/figures and text of a manuscript. Please see pages 4 and 5.

  • In the introduction, some more information concerning mutuality and its role in schizophrenia and recovery form could be provided

Reply: Thanks for the reviewer’s valuable suggestion. We have added more information related to mutuality in schizophrenia and recovery in the introduction. Please see page 2.

Greater mutuality was associated with higher HRQoL in outpatients suffering from schizophrenia [16] and caregivers of inpatients diagnosed with schizophrenia [17].

References:

  1. Hsiao, C. Y.; Hsieh, M. H.; Tseng, C. J.; Chien, S. H.; Chang, C. C. Quality of life of individuals with schizophrenia living in the community: Relationship to socio-

demographic, clinical, and psychosocial characteristics. J. Clin. Nurs. 2012, 21, 2367-2376.

  1. Hsiao, C. Y.; Lu, H. L.; Tsai, Y. F. Caregiver burden and health-related quality of life among primary family caregivers of individuals with schizophrenia: A cross-sectional study. Lif. Res. 2020, 29, 2745-2757.
  • It is not clear from the presentation why as participants those who reside in the inpatient facility were selected and for how long they resided in such facility before recruitment-for such selection of participants it is problematic in which context they put their mutuality and relationship with caregivers, what is more their HRQoL might be affected by their “inpatient status”. Such situation is not addressed by the authors.

Reply: Thanks for the reviewer’s valuable suggestions.

  1. To address the important issues of research on people in inpatient psychiatric rehabilitation facilities, we added more information to the introduction. Please pages 1-2.

Recovery is recognized as an ultimate goal for mental healthcare services [1]. Worldwide the disability-adjusted life years of schizophrenia increased by 62.46% in 2017 compared to 1990 [2]. Inpatient psychiatric rehabilitation, as an interim pathway, provides recovery-oriented care to enable people who are disabled by mental disorders to live satisfactory lives [3, 4].

   Considering the literature on patient-caregiver dyads in medical diseases, one would anticipate that a reciprocal influence exists within the dyadic context of recovery for patients with mental disorders such as schizophrenia. However, there is a gap in the literature for the dyadic relationship of mutuality with HRQoL in individuals suffering from schizophrenia and their caregivers. This gap poses a major barrier to identifying risks for patient- caregiver dyads, which prevents an advanced understanding of the dyadic nature and the development of strategies for improving HRQoL within patient-caregiver dyads. Further, research on patients with persistent and complicated demands in inpatient psychiatric rehabilitation facilities is limited [4]. Therefore, the aims of this study were to investigate the dyadic relationship between mutuality and HRQoL in patients with schizophrenia and caregivers. It was hypothesized that higher levels of mutuality would be related to better HRQoL in patient-caregiver dyads confronting schizophrenia.

  1. We added more information about inpatient psychiatric rehabilitation and inclusion criteria of patients in the study design and sampling. Please see page 3.

This was a cross-sectional, correlational study following the Strengthening the Reporting of Observational Studies in Epidemiology (STROBE) standards [23]. A convenience sample of 133 dyads of patients with schizophrenia and caregivers was recruited from inpatient psychiatric rehabilitation facilities of two psychiatric hospitals in Taiwan. Inpatient psychiatric rehabilitation facilities with recovery-oriented practice provide 24-hour mental health support for patients who require longer inpatient treatment and are not yet ready for successful community reintegration.

Inclusion criteria of patients were: (a) diagnosed with schizophrenia based on the criteria of the Diagnostic and Statistical Manual of Mental Disorders, Fifth Edition (DSM-5) [24]; (b) residing in an inpatient psychiatric rehabilitation facility for at least six months at recruitment; (c) age of 20 years or older; and (d) capable of understanding or communicating in Taiwanese or Mandarin.

  1. Given the length of admission to inpatient psychiatric rehabilitation facilities may affect the mutuality and HRQoL in patient-caregiver dyads, we added this into the limitations. Please see page 7.  

Given a cross-sectional design, findings from the current study preclude understanding the casual relationships between mutuality and HRQoL for participant dyads during inpatient psychiatric rehabilitation services. Longitudinal research is recommended to confirm our findings. Other potential factors, such as length of admission in inpatient psychiatric rehabilitation facilities at recruitment or treatment duration of patients relevant to HRQoL, warrant further investigation. Lastly, results found within our group may not be generalized to other clinical settings, such as outpatient clinics.

  • In the presentation of the results repetition of data in tables/figures and text could be omitted.

Reply: Thanks for the reviewer’s recommendations. We have revised the results. Please see pages 4 and 5.

Characteristics of patient-caregiver dyads are presented in Table 1. As displayed in Table 2, patients reported a higher mean score for Mutuality Scale than that of caregivers (t = 5.6, p < .001). Mean scores of HRQoL and its domains in patients were significantly lower than those in caregivers, with the exception of the psychological domain. Additionally, significant correlations were identified between patients’ mutuality and caregivers’ mutuality (r = 0.59, p < .001) as well as mean scores of HRQoL and its domains for patients and caregivers, with coefficients ranging from 0.26 to 0.56.

Figure 1 depicts that patients’ and caregivers’ greater mutuality were significantly related to better HRQoL for themselves (actor effects) and their partners (i.e., caregivers and patients) (partner effects). 

As presented in Figure 2, the only significant partner sole effects of mutuality on physical health for patients and caregivers were found. Actor effects of patients’ and caregivers’ mutuality on their own psychological health (ß=0.28, p < .01 for patients; ß=0.36, p < .001 for caregivers) and a partner effect of caregivers’ mutuality on patients’ psychological health (ß=0.21, p < .05) were detected. Actor and partner effects of mutuality on social relationships for patients and caregivers were significant. Actor effects of patients’ mutuality and caregivers’ mutuality on their environment (ß=0.37, p < .001 for patients; ß=0.43, p < .001 for caregivers), and a partner effect of mutuality on environment from caregivers to patients (ß=0.3, p < .001) was found.

Round 2

Reviewer 3 Report

I've looked at the Authors' response and the corrected part of their manuscript. I think that the changes that have been made improve the manuscript and made it more clear. Thus the new version of the text can be accepted for publication in IJERPH. With best regards